# OpenReview forum: "Differentiable Verification for Safe Reinforcement Learning in Verifiable Code Synthesis"
_ICLR.cc/2026/Conference — Submitted to ICLR 2026_

### Official Review · Reviewer_xYfA · 2025-10-21

**Soundness:** 1
**Presentation:** 1
**Contribution:** 1
**Rating:** 0
**Confidence:** 5

**Summary:**

This paper combines reinforcement learning with differentiable approximations of type constraints and memory safety specifications to generate code that can be type-checked and proven to adhere to memory specifications using an SMT-based verifier. The differentiable approximation is partially designed manually and partially learned. This approximation is used both during reinforcement learning and when generating programs. It also decomposes generating programs into generating a syntax tree first that is subsequently filled in to produce the concrete code.

**Strengths:**

The paper aims to address a relevant problem (generating programs under formal specifications) by employing the promising approach of constraining generation based on specifications. It cites several relevant works and provides experiments comparing the approach to several baselines.

**Weaknesses:**

The primary issue with this paper is its writing style. I was unable to follow most of the paper and cannot confidently say that I understood how the different ideas presented in the paper interact to generate code. I might be wrong about the points below since I was unable to follow the paper. However, I think the writing style alone makes it impossible to accept this paper. Below, I will briefly mention several issues besides the writing style and then provide recommendations for improving the presentation.

1. As far as I can see, each of the techniques the paper uses has already been applied in earlier works cited in Section 2. As such, the paper lacks novelty.
2. According to Section 4.3, training the policy requires running the program verifier several times per parameter update in the inner optimisation loop. Since running the verifier is costly, this makes training prohibitively expensive.
3. The experiments section does not compare to the state-of-the-art approaches discussed in Section 2. The most recent work compared against is from 2019.

**Questions:**

I will list suggestions for improving the writing here.

1. Most importantly, describe how the different techniques you introduce play together to form the overall algorithm. Do this both for learning and for generation.
2. Describe what you do differently from the works in Section 2.3.
3. Properly define the problem you are trying to solve. Equation (1) defines a function. It is not a constraint satisfaction problem in this form.
4. Provide concrete examples for the specifications you are using.
5. Use consistent notation. For example, in Equation (6), R has one parameter, but in Equation (9), it has three.
6. Introduce the terms you use. For example, what do you mean by "surrogate drift"?
7. Write full sentences and reduce the number of enumerations and itemizations.
8. Check spelling and grammar. For example, "bunkmarks" in the conclusion; missing verb in "while generality and specificity" in the introduction. An LLM can flag this for you.

---

### Official Review · Reviewer_teG9 · 2025-10-26

**Soundness:** 1
**Presentation:** 1
**Contribution:** 1
**Rating:** 2
**Confidence:** 5

**Summary:**

This work proposed a novel framework for safe RL from program synthesis perspective to enable the formal verifiability and bi-level end-to-end training structure. However, the work itself is very uncompleted in both writing and experimental results. Besides, more technical details are needed to validate the algorithmic design.

**Strengths:**

The framework author proposed is interesting and novel in the field. However, the work itself is very uncompleted, thus no further strengths can be analyzed there.

**Weaknesses:**

1, Authors missing some literature context in introduction: works as [1,2] also propose verification/optimization gradient feedback to enable safe RL with control barrier function with formal guarantees in a end-to-end bilevel optimization approaches.
2, The related work section is poorly written, missing the entirely safety filter based methods, such as CBF, HJB, etc. Noted that even though [1] has been mentioned, but in the wrong context. This citation and many others are hallucinated with incorrect conference, year, and author orders.
3, Besides, each subsection doesn't correspond to correlated content for section 2.
4, The whole hierarchical program generation part is every unclear, more details are needed in this subsection for understanding.
5, Line 178-180 needs more explanation, and how the approximate verification mechanism works to provide formal guarantee.
6, The introduction of surrogate verification is only to approximate SMT results, which reviewer believe has quite limited computing efficiency problem, makes the overall framework overly complicated.
7, Convergence guarantee needed to be given for equation 8 and 9 to state the somewhat formal verifiability.
8, The illustrations on 4.4, 4.5, and 4.6 are very vague. More details are needed to validate effectiveness of the work.
9, For experiment, the benchmark tasks need more explanation and introduction. How the coding tasks can be framed into a MDP.
10, If transformer is used as policy network, total number of parameters are needed. And comparison with current foundation models are also suggested to provide.
11, The ablation studies need serious rewriting, so does the whole experiment section.


[1], Wang, Yixuan, et al. "Joint differentiable optimization and verification for certified reinforcement learning." Proceedings of the ACM/IEEE 14th International Conference on Cyber-Physical Systems (with CPS-IoT Week 2023). 2023.
[2], Choi, Jason, et al. "Reinforcement learning for safety-critical control under model uncertainty, using control lyapunov functions and control barrier functions." arXiv preprint arXiv:2004.07584 (2020).

**Questions:**

1, line 16-17 'inefficiencies and mismatches between the generated code and safety guarantees' this statement is confusing within the abstract context
2, What is type safety mentioned in line 116? And what is memory safety in line 120? If those are constraints in program synthesis how can they related with safe RL context and formulation?
3, For equation 7, the second term why the derivative is with respect to \theta and how to calculate this gradient?
4, Is current RL method on foundation model? If it is, please change the formulation adaptively. Besides, how authors adapt some of the baselines from traditional safe RL method to safe RL on foundation models are also needed.

---

### Official Review · Reviewer_UBkr · 2025-10-31

**Soundness:** 1
**Presentation:** 1
**Contribution:** 1
**Rating:** 0
**Confidence:** 2

**Summary:**

The paper introduces a policy-optimization framework for safe reinforcement learning (RL) with code synthesis.

**Strengths:**

Integrating differentiable verification into RL has the potential to ensure safety without sacrificing efficiency.

**Weaknesses:**

I believe this paper was submitted prematurely and currently resembles an outline. More detailed explanations and discussion are needed in each section for the paper to be properly reviewed.

**Questions:**

What is your plan for improving this paper?

---

### Official Review · Reviewer_w6et · 2025-10-31

**Soundness:** 2
**Presentation:** 1
**Contribution:** 2
**Rating:** 2
**Confidence:** 4

**Summary:**

this paper proposes a novel end-to-end framework integrating differentiable formal verification into safe reinforcement learning  for verifiable code synthesis, addressing inefficiencies of traditional methods that treat verification as a post-hoc filter or black-box reward.

Traditional approaches decouple verification from RL policy optimization, leading to trial-and-error in generating safe code and a disconnect between continuous RL dynamics and discrete verification. The framework solves this by modeling verification constraints as differentiable functions  to enable gradient-based joint optimization of code generation and safety.

Its core components include a differentiable verification layer, a hierarchical policy , bilevel optimization , and periodic hard-constraint injection to avoid surrogate drift. It also uses modular synthesis to decompose complex programs into verified submodules.

Experiments on algorithmic, system programming, and DSL tasks show the framework outperforms baselines: 95.8% Verification Success Rate, 74.6% Functional Correctness , 5× faster verification than post-hoc methods, and higher Synthesis Quality. Ablation studies confirm key components boost VSR.

Limitations include surrogate approximation gaps for complex properties and reward-hacking risks. Future applications include autonomous systems and smart contracts. The work advances deployable AI programming assistants with provable safety.

**Strengths:**

1. The work pioneers integrating differentiable formal verification into safe RL for code synthesis, solving the inefficiency of traditional post-hoc verification or black-box rewards by enabling gradient-based joint optimization of code generation and safety.

2. the paper designs effective relaxations  to model discrete verification as continuous functions, preserving semantics while enabling gradient flow.

3. The authors address safety-critical domains  including autonomous systems, smart contracts, detects 89% of smart contract reentrancy vulnerabilities , and balances training overhead with deployment efficiency.

**Weaknesses:**

1.  The quality of the differentiable verification surrogate (\(\tilde{V}\)) heavily depends on feature representations. For complex properties e.g., quantifiers, nonlinear arithmetic, intricate loop invariants, approximation errors persist—only 78% of verifiable loop invariant cases are captured in benchmarks. This limits applicability to safety-critical domains needing rigorous verification of such properties.

2. The two-level policy  introduces compounding errors. Mistakes in high-level AST planning are amplified during low-level token instantiation, as the differentiable verification layer cannot fully correct early structural flaws, reducing robustness for long, complex programs.

3. Poor Presentation: The paper requires careful proofreading before publication, as it contains several issues that hinder clarity and adherence to standard research paper conventions.
First, there are numerous one-sentence paragraphs throughout the text, which fail to provide sufficient context or explanation. For instance, in Section 2 (Related Work), short, isolated sentences like “Differentiable approximations of formal methods have been a promising direction to bridge this gap” are left underdeveloped—they do not elaborate on why this direction is promising or how it relates to the paper’s contributions, forcing readers to infer connections.
Second, Section 5 (Experimental Evaluation) deviates significantly from typical research paper structure and style. Key subsections (e.g., 5.4 Case Studies) rely on fragmented, bullet-point-like statements without cohesive narrative flow. For example, the memory safety case study lists observations (“Insert bounds checks (94% of cases)”) but lacks explanatory text linking these findings to the framework’s core mechanisms (e.g., how the differentiable verification layer enables learning bounds-check behavior). Additionally, critical details (e.g., how unit tests for Functional Correctness (FC) were designed, or the criteria for selecting benchmark tasks) are scattered across short, disjointed paragraphs, making it difficult to follow the experimental logic or replicate the results.

**Questions:**

1. The paper mentions that verification surrogates struggle with complex properties, capturing only 78% of verifiable loop invariant cases. Have you explored specific feature engineering strategies  to narrow this approximation gap, and what performance improvements might such strategies bring?


2. The hierarchical policy’s multi-step generation leads to compounding errors. Did you test alternative policy structures to mitigate this issue, and how do these structures compare in terms of verification success rate and code generation quality?

2. For real-world deployment , your method detects 89% of reentrancy vulnerabilities—yet post-hoc tools still miss some. Have you analyzed why the differentiable verification layer fails to capture these remaining vulnerabilities, and do you plan to extend the framework to support additional formal logics to address such gaps?

---

### Meta-Review · Area_Chair_h3q9 · 2026-01-07

**Summary:**

The paper seems to be written by a LLM. This issue was raised unanimously by all reviewers (UBkr, xYfA, w6et) since the paper appears to be an unfinished draft rather than a complete submission. There were multiple hallucinated references as well. The writing and plots point strongly towards this hypothesis.

Therefore, I recommend rejecting the paper.

**Reviewer Concerns:**

No rebuttal was posted.

**Reviewer Scores:**

No rebuttal was posted.

---

### Decision · Program_Chairs · 2026-01-26

Reject